# Genome-Wide Investigation and Expression Analysis of the *Nitraria sibirica* Pall. CIPK Gene Family

**DOI:** 10.3390/ijms231911599

**Published:** 2022-09-30

**Authors:** Liming Zhu, Hao Fang, Ziming Lian, Jingbo Zhang, Xinle Li, Jisen Shi, Lu Lu, Ye Lu, Jinhui Chen, Tielong Cheng

**Affiliations:** 1Key Laboratory of Forest Genetics & Biotechnology of Ministry of Education of China, Co-Innovation Center for Sustainable Forestry in Southern China, Nanjing Forestry University, Nanjing 210037, China; 2College of Biology and the Environment, Nanjing Forestry University, Nanjing 210037, China; 3Experimental Center of Desert Forestry, Chinese Academy of Forestry, Dengkou, Bayannur 015200, China

**Keywords:** *CIPKs*, abiotic stress, *Nitraria sibirica* pall., expression pattern

## Abstract

The calcineurin B-like-interacting protein kinase (CIPK) protein family plays a key role in the plant calcium ion-mediated signal transduction pathway, which regulates a plant’s response to abiotic stress. *Nitraria sibirica* pall. (*N. sibirica*) is a halophyte with a strong tolerance for high salt environments, yet how it is able to deal with salt stress on a molecular level is still unknown. Due to their function as described in other plant species, *CIPK* genes are prime candidates for a role in salt stress signaling in *N. sibirica*. In this study, we identified and analyzed the phylogenetic makeup and gene expression of the *N. sibirica CIPK* gene family. A total of 14 *CIPKs* were identified from the *N. sibirica* genome and were clustered into seven groups based on their phylogeny. The promoters of *NsCIPK* genes contained multiple elements involved in hormonal and stress response. Synteny analysis identified a total of three pairs of synteny relationships between *NsCIPK* genes. Each gene showed its own specific expression pattern across different tissues, with the overall expression of *CIPK6* being the lowest, and that of *CIPK20* being the highest. Almost all CIPK genes tended to respond to salt, drought, and cold stress, but with different sensitivity levels. In this study, we have provided a general description of the *NsCIPK* gene family and its expression, which will be of great significance for further understanding of the *NsCIPK* gene family function.

## 1. Introduction

Currently, the global ecological environment is progressively deteriorating. Drought and an increasing saline-alkali concentration lead to soil deterioration. More than 932.2 M ha of soil around the world have been affected by saltification and alkalization leading up to the year 2016 [1]. Most plants are sessile, and major changes in their surrounding environment will dramatically affect their growth and development. Climate change and environmental degradation lead to severe abiotic stress in plants [2,3], resulting in significant damage to plant life, both affecting their survival and limiting their global distribution.

In order to survive in harsh environments, plants have evolved a multitude of molecular mechanisms to adapt their physiology, the Ca^2+^ signaling pathway being one of the most important pathways involved in their response to abiotic stress [4]. When plants suffer from abiotic stress, the Ca^2+^ concentration in plants increases. Ca^2+^ signal receptors sense the change in Ca^2+^ concentration and activate related protective mechanisms by regulating downstream gene expression to avoid sustained cellular damage. Among the key Ca^2+^ receptors are Calmodulin (CaM) [5], Calmodulin-like protein (CMLs) [6], Calcium-dependent protein kinases (CDPKs) [7], and calcineurin B-like protein (CBLs) [8]. CBLs physically interact with CBL-Interacting Protein Kinases (CIPKs) to facilitate signal transduction, which is one of the Ca^2+^ signal transduction pathways that has been studied extensively [9]. For example, *CBL4* and *CBL10*-*CIPK24* (*SOS2*)-*NHX7* (*SOS1*) interaction networks jointly regulate the salt stress response pathway [10,11]. *CBL1* and *CIPK7* are involved in cold stress response [12]. *CIPK14* interacts with *CBL1* to regulate the response to drought stress in pigeon pea [13]. In addition, *CIPK* genes are also involved in the regulation of responses to further stresses such as high pH [14], low potassium [15] and abscisic acid (ABA) stress [16]. Collectively, CIPKs play an important role in plant abiotic stress response pathways.

The *CIPK* genes encode serine/ threonine-protein kinases, which contain two key domains; the N-terminal catalytic/kinase domain and the C-terminal NAF/FISL domain. The N-terminal kinase region can be phosphorylated to enhance kinase activity, but the C-terminal NAF/FISL inhibits kinase activity [17]. However, the NAF/FISL domain of CIPK protein kinases can be converted into a protein kinase activator by physically binding with the *CBL* gene [18]. Because of their demonstrated importance as abiotic stress mitigators in plants, *CIPK* genes have been identified and studied in multiple plant species. For example, 26, 34, and 43 *CIPK* genes were identified in *Arabidopsis*, rice, and maize, respectively [19,20,21,22].

*N. sibirica* is a shrub plant that is placed in the *Nitraria Linnaeus* genus and is mainly distributed in saline-alkali arid areas as it can withstand very severe salt and drought damage. *N. sibirica*’s saline-alkali tolerance gives it the potential to be a pioneer species in saline-alkali soil remediation. However, the molecular mechanisms underlying the tolerance of *N. sibirica* to adverse environmental conditions such as drought and salinity have not yet been elucidated, and whether CIPKs participate in such abiotic stress responses is unclear as well. Therefore, in this study, we characterized the *N. sibirica CIPK* gene family, analyzing their expression patterns in response to salt, drought, and cold treatments. We identified a total of 14 *CIPKs* from the *N. sibirica* genome. Their expression showed distinct responses to salt, drought, and cold stress, indicating that *NsCIPKs* may play a role in the molecular response to these abiotic stresses in *N. sibirica*. These results underscore the potential of *NsCIPKs* as antibiotic stress regulators in *N. sibirica* and serve as a solid basis from which to further study their function.

## 2. Result

### 2.1. Identification and Chromosomal Location of the N. sibirica CIPK Gene Family

In order to identify *CIPK* genes in *N. sibirica*, we used HMMER software to scan the *N. sibirica* genome library using both the CIPK kinase and NAF/FLSL domains. We selected as candidate genes those genes that gave a hit using either domain. In addition, we used a BLASTP algorithm to search for potential NsCIPKs, taking AtCIPKs as input sequence. The results from both the HMMER and BLASTP search were combined, and we used the SMART database and the conserved domain database to further determine whether candidate genes were indeed *NsCIPKs*. In total, 14 *CIPK* genes were identified in *N. sibirica* (Appendix A). To identify the distribution of these identified CIPKs across the whole genome, we mapped them to the chromosome level according to their ID numbers and genome data. We were able to map 12 *NsCIPK* genes directly to chromosomes, which were CHR3, CHR6, CHR7, CHR9, CHR10, and CHR11, respectively, while *NsCIPK1-1* and *NsCIPK1-2* were mapped to sequence scaffolds (scaffold80 and scaffold97) (Appendix A). Then, we performed all against all blastp searches between the identified candidate NsCIPKs and AtCIPKs. Based on their sequence homology to AtCIPKs, the NsCIPK genes were named as *NsCIPK1-1* to *NsCIPK23,* respectively. 

We then continued to further characterize the NsCIPK proteins by determining the number of amino acids and relative molecular weight (MW) (Table 1). The number of amino acids in the *N. sibirica* CIPK proteins lies between 359 to 583, with the MW varying from 39.04 kDa to 64.42 kDa, and the PI index is between 5.7 to 9.33. We then performed transmembrane structure analysis and found that only NsCIPK12 has a putative transmembrane domain, while the other 13 NsCIPKs do not (Appendix A). We used Cell-PLOc2.0 software to predict subcellular localization and found that all NsCIPK proteins are potentially localized to the cytoplasm. Furthermore, NsCIPK1-2, NsCIPK11, NsCIPK15, and NsCIPK23 are predicted to be localized to both the cytoplasm and the nucleus (Table 1).

### 2.2. Phylogeny and Synteny Analysis of the CIPK Gene Family

To elucidate the evolutionary relationship of the *NsCIPK* gene family with other species, we used the amino acid sequences of AtCIPKs, OsCIPKs, and NsCIPKs (Appendix A) to construct an unrooted phylogenetic tree. As shown in Figure 1, a total of 73 CIPKs could be divided into 7 groups. Among them, the largest is group G which contains 19 members, while groups A, D, and E have the smallest distribution with four members. The remaining groups B, C, and F have 16, 16, and 10 members, respectively (Figure 1). These results indicate that the functions of CIPKs may have diverged during evolution. We found no *CIPK* gene in groups A and E, which may indicate that CIPKs in this branch did not diverge or become lost during the evolution of *N. sibirica*.

We found that the number of *CIPKs* in *Arabidopsis* and rice is higher than that in *N. sibirica*. In order to verify whether some of the *CIPKs* have been lost during *N. sibirica* evolution, we used CIPKs from additional species (Appendix A) to draw a more extensive phylogenetic tree (Appendix A). This tree showed that, although *N. sibirica* has only 14 CIPKs, they are distributed on almost every branch of the phylogenetic tree. There is no clustering of NsCIPK on CIPK4 or 7, consistent with a phylogenetic tree with only three species (Figure 1). In addition, the synteny analysis between NsCIPKs also shows that there are only three collinear gene pairs (Figure 2), which indicates that the internal collinearity of CIPK is not strong. These results may indicate that the *CIPK* gene family is not expanded in *N. sibirica*. 

The presence of collinearity between genes in different species often indicates the similarity of gene functions. Therefore, we performed synteny analysis on the *CIPK* gene families of *N. sibirica*, *Arabidopsis,* and rice. A total of 15 pairs of syntenic genes were found between *Arabidopsis* and *N. sibirica*. However, there were only six syntenic genes between *N. sibirica* and rice. We found more syntenic gene pairs between *NsCIPKs* and *AtCIPKs* than between *NsCIPKs* and *OsCIPKs*, indicating that *NsCIPKs* and *AtCIPKs* are more closely related to each other than they are to *OsCIPKs* (Figure 2). 

To analyze whether the *CIPK* gene family had duplication events over a long evolutionary history, we searched for duplications using McScanX software [23]. However, we could not detect any segmental duplications in NsCIPKs. Next, we performed a synteny analysis to look for synteny between *NsCIPKs*. A total of three pairs of synteny relationships were identified in these *NsCIPKs*, namely, *NsCIPK14* and *NsCIPK11*, *NsCIPK11* and *NsCIPK3-1*, and *NsCIPK14* and *NsCIPK3-1*, respectively (Figure 2). This suggests that segment duplications may be the main expansion mode of the *CIPK* gene family.

### 2.3. NsCIPKs Contain Multiple Conserved Motifs

In order to further study the structure of these *CIPK* genes, we analyzed their conservative motifs, domain distribution and gene structure. Via domain analysis, we found that the kinase domains of NsCIPKs are all located near the N-terminus, while the NAF domains are all located at the C-terminus of the protein (Figure 3B and Appendix A). A total of 11 major motifs were identified in these NsCIPKs (Figure 3A and Appendix A). Among them, motif 1, motif 7, and motif 8 exist in all *NsCIPK* genes. The remaining motifs were present in most *CIPK* genes, which suggest that these CIPKs may have similar functions. Analysis of the *NsCIPK* gene structure showed that they can be divided into two categories, depending on whether introns are present or not (Figure 3C). *NsCIPK11*, *NsCIPK14,* and *NsCIPK15* have no intron structure, while the remaining NsCIPKs have one or multiple introns, with the total gene length remaining under 9000 bp.

### 2.4. NsCIPK Promoters Contain Multiple Stress Responsive cis Regulatory Elements

A *cis* regulatory element is a target nucleotide sequence that can be bound by different trans-acting factors, which are involved in the regulation of gene expression. We used PlantCare online tools to identify *cis* regulatory elements within the sequence region 2500 bp upstream of the start codon of each *NsCIPK* gene. On average, we could identify ten *cis* regulatory elements within these regions (Figure 4). These *cis* regulatory elements are mainly related to hormone response, salt, low temperature, and drought stress, which suggests that *NsCIPKs* may respond to hormones and abiotic stress.

### 2.5. NsCIPK Gene Expression Analysis across Different N. sibirica Tissues

To explore the expression of *NsCIPKs* in different tissues, we collected the root, stem, and leaves of two-month-old seedlings of *N. sibirica*, isolated RNA, and performed qRT-PCR. Since the mRNA sequence of *NsCIPK1-1* is highly similar to *NsCIPK1-2*, no specific primers could be designed. Therefore, *NsCIPK1-1* and *NsCIPK1-2* are collectively referred as *NsCIPK1*. We designed customized qRT-PCR primers for each gene (Appendix A), and the result showed that there were clear differences between the expression levels of individual *NsCIPK* genes in different tissues (Figure 5); *NsCIPK12* shows the highest expression in the root, while *NsCIPK20* is the most abundant in the stem and leaf. In general, *CIPK3-2*, *CIPK12*, *CIPK20*, *CIPK21*, and *CIPK23* show comparatively high expression levels in all tissues analyzed, while the rest of the genes showed a lower expression level. Notably, the expression of *CIPK6* in roots, stems, and leaves was at an extremely low level compared with other CIPK genes (about 1/1000 of the *CIPK1* expression level). These results suggest that there are dedicated CIPK genes functioning in different tissues while some CIPK genes may only act in the event of special environmental conditions.

### 2.6. Expression Patterns of NsCIPK Genes in Response to Different Abiotic Stresses

Studies in other species have found that CIPK families generally respond to abiotic stress [24]. To explore whether *NsCIPKs* respond to abiotic stress as well, we used treatments with 300 mmol/L NaCl, 20% PEG6000 and 4 °C to simulate salt, drought and cold stress, respectively, after which we detected the expression level of the *NsCIPK* gene family by quantitative real-time PCR. As shown in Figure 6, for salt stress, we found that most of the CIPK genes were induced to express, although the period of expression is different. However, *NsCIPK3-1* and *NsCIPK12* were down-regulated in some periods, and only *NsCIPK15* did not show an obvious upward or downward trend. For drought stress, the relative expression levels of NsCIPK1-1, *NsCIPK3-1*, *NsCIPK3-2*, *NsCIPK6*, *NsCIPK11*, *NsCIPK12*, *NsCIPK14*, and *NsCIPK20* increased in different degrees, indicating that they may play a role in drought stress. In contrast, the rest of the CIPK genes showed a downward trend. In cold stress, most NsCIPKs increased and changed little, but only *NsCIPK8* expression tended to decrease during stress. These analyses indicated that different NsCIPK had different sensitivities to salt, drought, or cold stress environments, but in general, *NsCIPKs* responded to abiotic stress processes such as salt, drought, and cold stress, suggesting that they may play a key role in these abiotic stress processes.

## 3. Discussion

Salinization and aridification are becoming more and more severe, limiting the further development of global agriculture and affecting food output [25]. Phytoremediation has always been an effective measure for soil remediation because it is a fully natural and low-cost approach. Salt-alkali tolerant and drought-tolerant plants are especially suited for this purpose. *N. sibirica* has excellent tolerance for drought and saline-alkali environments and is widely distributed in the east of Eurasia, making it an ideal plant for the treatment of salinization and alkalinity. 

The calcium signaling pathway is one of the key molecular pathways involved in the biological response to an adverse environment and includes multiple typical response pathways such as the CAM, CDPK, and CBL-CIPK pathways. CIPK genes regulate a plant’s response to Ca^2+^ by binding the CBL gene. CIPK binds CBL through its own C-terminal NAF domain, making the NAF domain indispensable for CIPK function [17]. A NAF domain was found in all the *CIPK* genes identified in this study (Appendix A), indicating that all these identified *CIPK* genes have the potential to interact with the CBL gene. CBL-CIPK is a unique pathway in plants and plays an important role in plant stress resistance, such as resistance to cold, drought, and salt stress. For example, ScCBL-ScCIPK signaling network regulates the response process of *Solanum commersonii* to cold stress [26]; ScCIPKs were involved in the response of cotton to drought stress [27]; TaCIPKs were responsive to salt stress in *Triticum aestivum* [28]. In this study, the *NsCIPK* genes were systematically analyzed at a genome-wide level in *N. sibirica*, which will be beneficial for further understanding the function of the CIPKs and the molecular mechanisms underlying salt tolerance and drought resistance in *N. sibirica*. 

The number of *CIPK* gene family members often varies between plant species; in this study, fourteen *NsCIPK* genes have been identified, which is relatively low compared to the number of *CIPK* genes in *Arabidopsis* (26), *Oryza sativa* (34), *Vitis vinifera* (20), and *Zea mays* (36). From the phylogenetic tree showing the interrelatedness between NsCIPK, AtCIPKs, and OsCIPKs, we found there was no clustering of *NsCIPKs* on *CIPK4* or 7. This may indicate that these *CIPK* genes were lost during *N. sibirica* evolution. To further investigate whether CIPKs were lost during *N. sibirica*’s long evolutionary history, we constructed phylogenetic trees with more species added (Appendix A) and found that NsCIPKs were distributed in almost every branch, which may indicate that a lack of extensive expansion of NsCIPKs may be the reason for the low number of *CIPKs* in *N. sibirica*. 

Genes tend to have tissue-specific expression, related to their function. In our expression analysis, we found that the genes *NsCIPK3-2*, *NsCIPK12*, *NsCIPK14,* and *NsCIPK20* are expressed in a tissue specific manner. By comparison, *NsCIPK3-1*, *NsCIPK6*, *NsCIPK9,* and *NsCIPK15* are always lowly expressed (Figure 5). The level of basal gene expression does not necessarily indicate whether a gene is functional or not; rather, it is the rise or fall of expression in a given environment that may determine the potential function of a gene. Some CIPKs, such as *NsCIPK1* and *NsCIPK6*, are expressed at a very low level during normal growth, but are strongly induced during stress, suggesting that these genes are likely transducing Ca^2+^ signals when plants are stressed.

We also noticed that some *CIPK* genes respond to multiple abiotic stresses simultaneously, while others only respond to one specific abiotic stress. These results suggest that there may be functional differentiation between *CIPK* genes. 

ABA has been shown to play an important role in plant tolerance to abiotic stress [29,30,31]. Several studies have found that *CIPK* genes play an important role in the ABA signaling response. For example, *AtCIPK1* plays a role in the ABA response pathways in *Arabidopsis,* as mutation of *AtCIPK1* results in an abnormal ABA response and affects plant tolerance to abiotic stress [32]. There are also studies showing that *AtCIPK3* is involved in cold stress and ABA response. *AtCIPK6* plays a role in plant tolerance to salt or osmotic stress [33,34]. In this study, we found that *NsCIPK1*, *NsCIPK3-1,* and *NsCIPK3-2* expression was significantly induced by salt, drought, and cold stress, suggesting that a similar response mechanism may exist in *N. sibirica*. *NsCIPK6* has similar expression trends to *NsCIPK3*, although *CIPK6* has not been reported to regulate cold stress, even though it has been reported that *CIPK6* can be induced by cold stress [35], which is consistent with what we found in this study. This suggests that *NsCIPK6* may have a function in plant cold stress response, motivating further functional studies. In addition, even *NsCIPK* genes with less pronounced differential expression during abiotic stress did contain ABA responsive elements (ABRE) in their promoter region, which may indicate that ABA activates these *CIPKs* during abiotic stress in *N. sibirica*. It is important for us to further study the relationship of NsCIPKs and ABA in the process of abiotic stress response. The function of *NsCIPK* genes cannot be predicted only by their expression level and still needs to be verified by further experiments. It is of great significance to further study the function of *CIPK* genes in *N. sibirica* as they may have an important function in the molecular pathways underlying its profound abiotic stress resistance.

## 4. Materials and Methods

### 4.1. Plant Materials and Abiotic Stress Treatments

*N. sibirica* seeds were collected in Dengkou County, Inner Mongolia, China. Seeds were mixed with wet sand and placed at 4 °C to vernalize for two months. After the vernalized seeds germinated, they were planted in a soil mix of peat soil: perlite, with a ratio of 4:1. Plants were cultivated in the green-house under a 16 h-light/8 h dark- light cycle and 60% air humidity at 23 °C. After a growth period of around two months, the plant materials were used for expression analysis. For abiotic stress treatments, a 300 mM NaCl solution, 20% PEG 6000, and 4 °C were used to simulate salt, drought and cold stress, respectively. Plant materials were collected at 0, 2, 4, 8, 12, and 24 h after corresponding treatments and were quickly put into liquid nitrogen and transferred to −80 °C refrigerator for storage until RNA was extracted.

### 4.2. Identification of CIPK Genes in N. sibirica

*N. sibirica* genomic data were obtained from the unpublished complete genomic sequence that we obtained previously in our laboratory. *CIPK* gene sequences from *Arabidopsis* were extracted from the *Arabidopsis* genome (TAIR.10) according to their annotation ID. Subsequently, the Protein Kinase domain (PF00069) and NAF/FLSL (PF03822) domain from the Pfam database [36] (http://pfam.xfam.org/ (accessed on 21 May 2022)) were used to scan the putative protein sequence libraries of *N. sibirica* using HMMER software (v3.0) with an e-value of 10^−5^. Blastp software (v2.9) and SMART [37] (http://smart.embl-heidelberg.de/ (accessed on 21 May 2022)), and CDD search tools [38] (https://www.ncbi.nlm.nih.gov/Structure/bwrpsb/bwrpsb.cgi/ (accessed on 21 May 2022)) were then used to manually confirm the extracted sequences. Finally, these NsCIPKs were named according to their homology with AtCIPK proteins. The expasy tool [39] (https://prosite.expasy.org/ (accessed on 21 May 2022)) was used for the identification of the basic physical properties including MWs (molecular weights). The transmembrane structure was analyzed using the online analysis tool TMHMM2.0 [40] (https://services.healthtech.dtu.dk/service.php?TMHMM-2.0 (accessed on 21 May 2022)). The subcellular localization of NsCIPK proteins was predicted by using the online tool Cell-PLoc 2.0 [41] (http://www.csbio.sjtu.edu.cn/bioinf/Cell-PLoc-2/ (accessed on 21 May 2022)).

### 4.3. Motifs and Gene Structure Analysis

Conserved gene motifs were analyzed using MEME online tools [42] (https://meme-suite.org/meme/tools/meme/ (accessed on 21 May 2022)). Pfam sequence search (http://pfam.xfam.org/ (accessed on 21 May 2022)) was used to analyze the protein kinase domain and NAF/FLSL domain of predicted genes, and gene structure was analyzed by the GSDS [43] (http://gsds.cbi.pku.edu.cn/ (accessed on 21 May 2022)) online analysis tool.

### 4.4. Promoter cis Regulatory Element Prediction

Sequences 2500 bp in length located upstream of the coding region of putative genes were extracted for *cis* regulatory element analysis using PlantCare online tool [44] (http://bioinformatics.psb.ugent.be/webtools/plantcare/html/ (accessed on 21 May 2022)) and visualized by Tbtools software [45].

### 4.5. Chromosomal Location and Synteny Analysis

Chromosomal location was analyzed and visualized by Tbtools [45]. Criteria for gene duplication refer to the methods previously reported [23], and the result was visualized by Tbtools [42].

### 4.6. Phylogenetic and Multiple Alignment Analysis

The phylogeny of CIPKs from *N. sibirica*, rice, and *Arabidopsis* was analyzed using MEGA X v10.1.8 software (Temple, Philadelphia, PA, USA). The Muscle method was used to align the CIPK amino acid sequences [46], and the NJ (Neighbor-joining) method was used to construct the phylogenetic tree. The bootstrap was repeated 1000 times. The phylogenetic tree was visualized using Evolview software [47]. DNAMAN v9.0 (Lynnon Corporation, San Ramon, CA, USA) software was used for multi-fragment alignment of amino acid sequences.

### 4.7. Expression Analysis of CIPK Genes in N. sibirica

Total RNA was extracted from plant materials using an RNA extraction kit (Promega, Shanghai, China). cDNA was synthesized using a reverse transcription kit (Vazyme, Nanjing, China). Quantitative real-time PCR (qRT-PCR) was performed using a Lightcyler 480II (Roche, Basel, Switzerland) and the AceQ qPCR SYBR Green Master Mix (Vazyme, Nanjing, China). Primers for qRT-PCR were designed using the NCBI online primer design tool [48] (https://www.ncbi.nlm.nih.gov/tools/primer-blast/ (accessed on 21 May 2022)). The primers used in this experiment are shown in Appendix A. Three biological replicates of each sample were used for quantitative real-time PCR. The relative expression level was calculated using the 2^−ΔΔCT^ method [49], and the graph was finally plotted using GraphPad Prism (v8.01).

## 5. Conclusions

In this study, we identified and analyzed the phylogenetic makeup and gene expression of the *N. sibirica CIPK* gene family. A total of 14 *CIPKs* were identified from the *N. sibirica* genome and were clustered into seven groups based on their phylogeny. The promoters of *NsCIPK* genes contained multiple elements involved in hormonal and stress response. Synteny analysis identified a total of three pairs of synteny relationships between *NsCIPK* genes. Each gene showed its own specific expression pattern across different tissues, with the overall expression of *CIPK6* being the lowest, and that of *CIPK20* being the highest. Almost all *CIPK* genes tended to respond to salt, drought, and cold stress, but with different sensitivity levels. In this study, we have provided a general description of the *NsCIPK* gene family and its expression, which will be of great significance for further understanding of the *NsCIPK* gene family function.

## Figures and Tables

**Figure 1 ijms-23-11599-f001:**
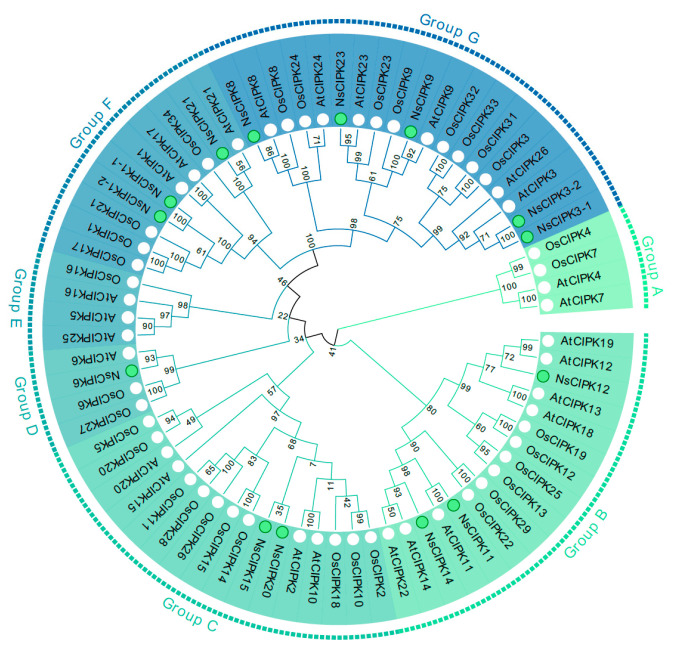
Phylogenetic relationships between CIPKs from *Arabidopsis*, rice, and *N. sibirica*. Genes marked with a green dot are *NsCIPKs*; those marked with a white dot are *AtCIPK* and *OsCIPK*.

**Figure 2 ijms-23-11599-f002:**
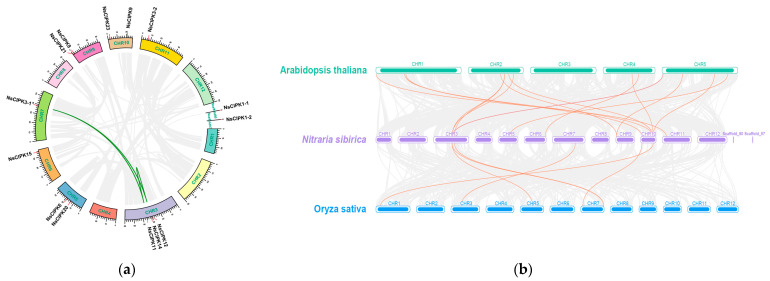
Genome-wide synteny analysis of CIPK gene family among *N. sibirica* and other three species: (**a**) inter-chromosomal relationships of CIPKs in *N. sibirica* (the links on the green curve indicate synteny relationships between genes); (**b**) synteny analyses between the CIPKs of *N. sibirica*, *Arabidopsis*, and Rice. The green, purple and blue chromosome level genomes belong to *Arabidopsis*, *N. sibirica*, and rice, respectively. The orange lines represent a syntenic relationship between two genes.

**Figure 3 ijms-23-11599-f003:**
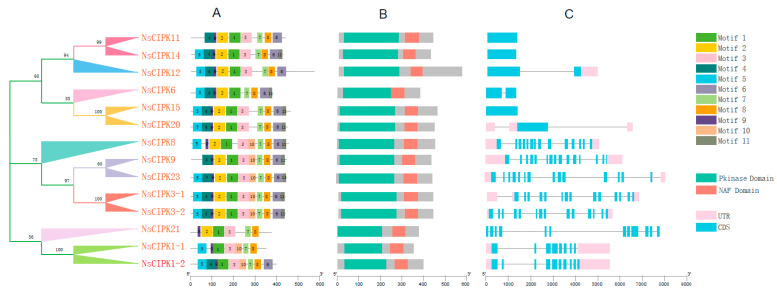
Conserved motifs, domain distribution and gene structure of N.sibirica CIPK genes. (**A**) Conserved motif distribution of NsCIPK genes. (**B**) Domain distribution of NsCIPK genes. (**C**) Gene structure of NsCIPK gene families.

**Figure 4 ijms-23-11599-f004:**
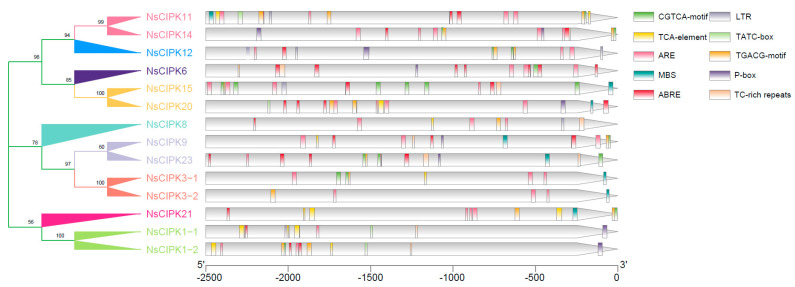
*cis* regulatory elements analysis of *N. sibirica* NsCIPK gene upstream regions. The colored squares represent the different *cis* regulatory elements.

**Figure 5 ijms-23-11599-f005:**
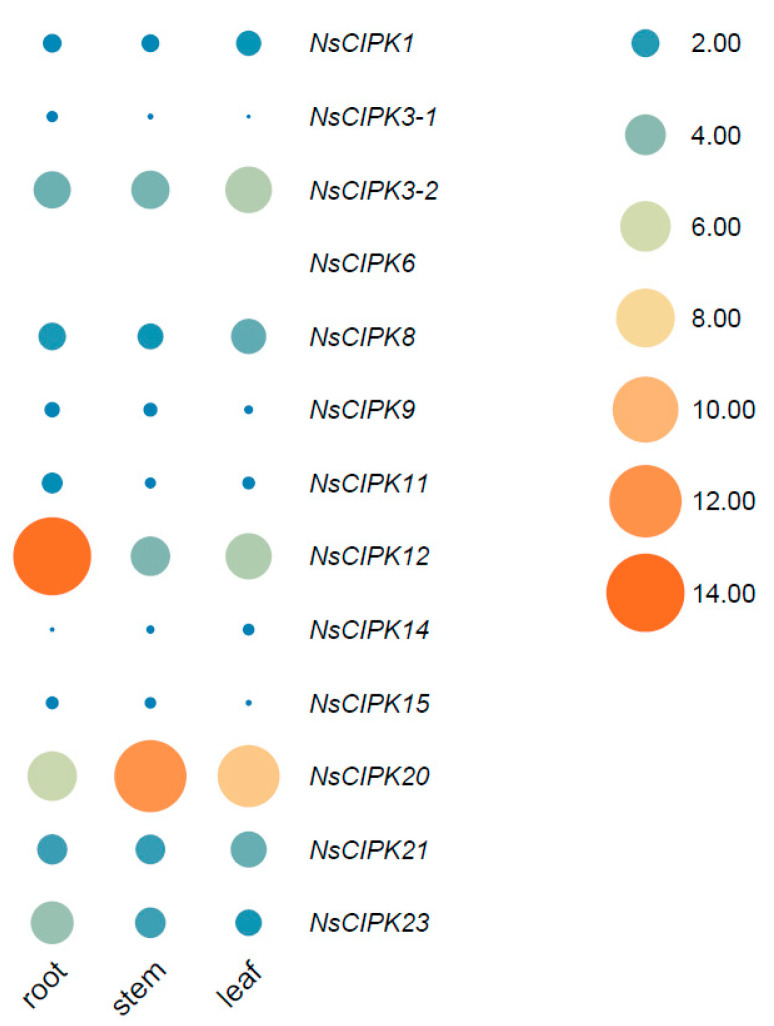
Comparative analysis of the relative expression of *CIPK* genes in different tissues of *N. sibirica*. Different circle sizes and colors indicate the relative expression size. The expression of *CIPK1* in the root was used as a reference for all relative quantification.

**Figure 6 ijms-23-11599-f006:**
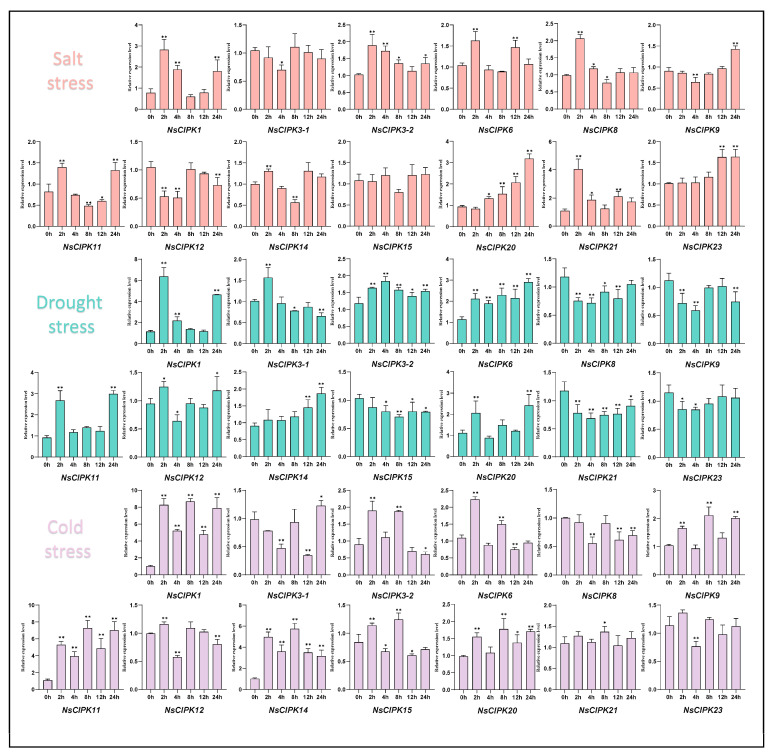
Expression analysis of *NsCIPKs* in response to salt, drought, and cold stress. A student’s *t*-test was used to analyze statistically significant differences (* *p* < 0.05, ** *p* < 0.01).

**Table 1 ijms-23-11599-t001:** Physicochemical properties of NsCIPK proteins.

Gene ID	Original ID	Locus	Protein Length (aa)	MW (kDa)	PI	Subcellular Localization
*NsCIPK1-1*	NISIscaf80G0007	Scaffold80	359	39.04	5.7	Cytoplasm
*NsCIPK1-2*	NISIscaf97G0002	Scaffold97	405	44.20	7.06	Cytoplasm, Nucleus
*NsCIPK3-1*	NISI07G1958	CHR7	447	50.08	8.4	Cytoplasm
*NsCIPK3-2*	NISI11G0565	CHR11	447	50.07	8.4	Cytoplasm
*NsCIPK6*	NISI05G0518	CHR5	390	43.20	8.97	Cytoplasm
*NsCIPK8*	NISI09G0120	CHR9	462	51.09	6.15	Cytoplasm
*NsCIPK9*	NISI10G0902	CHR10	451	50.54	8.94	Cytoplasm
*NsCIPK11*	NISI03G1369	CHR3	446	49.51	8.93	Cytoplasm, Nucleus
*NsCIPK12*	NISI03G1060	CHR3	583	64.42	9.33	Cytoplasm
*NsCIPK14*	NISI03G1063	CHR3	434	48.56	9.1	Cytoplasm
*NsCIPK15*	NISI06G1821	CHR6	471	52.54	8.80	Cytoplasm, Nucleus
*NsCIPK20*	NISI05G0308	CHR5	458	51.23	9.14	Cytoplasm
*NsCIPK21*	NISI09G0114	CHR9	383	43.48	9.29	Cytoplasm
*NsCIPK23*	NISI10G0044	CHR10	453	50.27	9.1	Cytoplasm, Nucleus

## Data Availability

The data in this study can be requested from the corresponding author.

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
