# Peer review of "Genome-Wide Investigation and Expression Analysis of the Nitraria sibirica Pall. CIPK Gene Family"

_ijms, 2022, doi:10.3390/ijms231911599_

Round 1
Reviewer 1 Report
Manuscript Liming Zhu, Hao Fang, Ziming Lian, Jingbo Zhang, Xinle Li, Jisen Shi, Lu Lu, Ye Lu, Jinhui Chen and Tielong Cheng «Genome-wide investigation and expression analysis of the Ni-2 traria sibirica Pall. CIPK gene family» is devoted to the actual problem: the mechanisms of resistance to abiotic factors (temperature, soil’s pH, drought) investigation. The authors note the problem of the annual augmentation of territories with stress environmental factors. The authors offer as one of the ways to counteract this phenomenon – rehabilitation of such territories with plants adapted to stress growing conditions.
Comments:
Line 56 Here the point of the first mention of ABA, it’s better to decoding the meaning of ABA here
Line 101 The values range given in the text does not correspond to the data in Table 1
Строка 200. Figure 7 described in the text is missing. Check the numbering of the illustrative material of the manuscript
Line 257 give only abbreviated name «ABA», decoding should be done at the first mention (line 56)
There is no mention of tables 2.3 of supplementary materials in the test of the manuscript
References sections contain double numbering
Author Response
Dear Reviewers:
Thank you for your decision and constructive comments on our manuscript. We have tried our best to improve the manuscript as suggested.
Comments:
Line 56 Here the point of the first mention of ABA, it’s better to decoding the meaning of ABA here
Response: Thank you for the comments. We added the meaning of ABA in the revised manuscript.
The modifications are listed in Line 56: abscisic acid (ABA).
Line 101 The values range given in the text does not correspond to the data in Table 1
Response: Thank you for the comments.It was our carelessness that led to this written error, we corrected in the revised manuscript.
The modifications are listed in Line 100: Change 9.69 to 9.33.
Line 200. Figure 7 described in the text is missing. Check the numbering of the illustrative material of the manuscript
Response: Thank you for the comments.We are sorry for our carelessness, we wanted to describe Figure 6 but mistakenly wrote Figure 7 instead, we corrected in the revised manuscript.
Line 257 give only abbreviated name «ABA», decoding should be done at the first mention (line 56)
Response: Thank you for the comments.We removed this line of decoding and explained it on line 57.
There is no mention of tables 2.3 of supplementary materials in the test of the manuscript
Response: Thank you for the comments. We have added descriptions of Supplementary Table 2 and 3 to the revised manuscript.
The modifications are listed in Line 119: we used CIPKs from additional species (Table S2) to draw a more extensive phylogenetic tree.
The modifications are listed in Line 182: We designed customized qRT-PCR primers for each gene (Table S3).
References sections contain double numbering
Response: Thank you for the comments. We removed duplicate numbers from the revised manuscript.
Reviewer 2 Report
The research design is appropriate to the final aim of the paper. The investigation of these interesting family goes from the function of each gene of the CIPK family (expression analysis in different tissues of N. sibirica and under abiotic stressors) to their structure even in the Cis regulatory elements. I just suggest to add some other interaction investigations and comparisons with other species (see Esposito, S., D'amelia, V., Carputo, D., & Aversano, R. (2019). Genes involved in stress signals: the CBLs-CIPKs network in cold tolerant Solanum commersonii. Biol plantarum, 63, 699-709.).
Author Response
The research design is appropriate to the final aim of the paper. The investigation of these interesting family goes from the function of each gene of the CIPK family (expression analysis in different tissues of N. sibirica and under abiotic stressors) to their structure even in the Cis regulatory elements. I just suggest to add some other interaction investigations and comparisons with other species (see Esposito, S., D'amelia, V., Carputo, D., & Aversano, R. (2019). Genes involved in stress signals: the CBLs-CIPKs network in cold tolerant Solanum commersonii. Biol plantarum, 63, 699-709.).
Response: Thank you for the comments. This advice is very useful, according to your suggestion we have added some other interaction investigations and comparisons with other species, which makes the discussion of the manuscript more specific.
The modifications are listed in line 231-234: For example, ScCBL-ScCIPK signalling network regulates the response process of Solanum commersonii to cold stress[27], ScCIPKs were involved in the response of cotton to drought stress[28], TaCIPKs were responsive to salt stress in Triticum aestivum[29].